# Identifying symptom cluster in cancer patients undergoing chemotherapy-in Vietnam: A cross-sectional study

**Huong Thi Xuan Hoang**[1]*, **Quyen Thi Le Le**[2], **Vi Pham Nhat Do**[3], **Anh Thi Hong Nguyen**[4], **Quang Vinh Bui**[2]

1 Nursing Faculty, Phenikaa University, Ha Dong District, Hanoi, Vietnam, 2 Hanoi Oncology Hospital, Hai Ba Trung District, Hanoi, Vietnam, 3 The Nethersole School of Nursing, Faculty of Medicine, The Chinese University of Hong Kong, Hong Kong, China, 4 Nursing Faculty, East Asia University of Technology, Hanoi, Vietnam

* huong.hoangthixuan@phenikaa-uni.edu.vn

**Data Availability Statement:** The data underlying the results presented in the study are available at https://doi.org/10.7910/DVN/1QCVHD.

## Abstract

### Background

Major cancer treatments can cause a wide range of unpleasant symptoms that burden oncology patients. Different symptom clusters (SC) among cancer patients have been reported in the literature. This study determined the prevalence of symptoms patients experience during chemotherapy treatment and identified symptom clusters among them.

### Methods

A cross-sectional study was conducted among 213 cancer patients undergoing chemotherapy in three large hospitals in Vietnam. Symptoms were measured by the Memorial Symptom Assessment Scale.

### Results

The most prevalent symptoms were lack of appetite (65.3%), difficulty sleeping (62.9%), dry mouth (57.7%), numbness (60.1%), hair loss (43.2%), change in the way food tastes (40.8%), and lack of energy (44.1%). Four symptom clusters were identified. Difficult concentration, pain, cough, and dizziness contributed to the first SC. The second one included *lack of energy, numbness, change the food taste, and lack of appetite. Dry mouth, nausea, feeling bloated, problems with urination, hair loss, and constipation* made up for the third SC. The final SC consisted of psychological symptoms, which were feeling *nervous, feeling drowsy, feeling sad, worrying, problems with sexual interest or activity and difficulty sleeping.*

### Conclusion

The study demonstrated that respondents experienced various symptoms and symptom clusters during chemotherapy. These findings can be used to develop clinical guidelines for symptom assessment and management in oncology patients for healthcare professionals.

**Funding:** The author(s) received no specific funding for this work.

**Competing interests:** The authors have declared that no competing interests exist.

## Introduction

Approximately one in six fatalities worldwide is attributable to cancer, making it a leading cause of death World Health Organization [1]. The global cancer burden is estimated to be 28.4 million cases in 2040 [2, 3]. Major treatments for cancer, such as surgery, chemotherapy, and radiotherapy, improve patient survival but can also have a substantial negative psychosocial and physical effect by exacerbating unpleasant symptoms that burden oncology patients [4–6]. Nearly half of cancer patients reported at least two symptoms throughout their illness, and on average, those who were undergoing chemotherapy reported at least ten symptoms that had a negative impact on their functional status and quality of life [7]. Hence, the concept of "symptom clusters" has been investigated. A symptom cluster has been defined as at least two symptoms that occurred and related to each other at a certain time, share a significant variance in their cluster, and may not have the same etiology [8, 9]. Different symptom clusters among cancer patients have been reported, such as *pain, fatigue, and sleep*; *anxiety and depression* [10] *fatigue, depression, and insomnia* [11, 12]; *pain, fatigue, and insomnia* [13–16] or *nausea and vomiting* [9]. However, the majority of the earlier research were conducted in a small sample size, and some studies only measured only 13 to 15 symptoms, while cancer patients can experience up to 32 symptoms [17]. In addition, previous studies explored symptom clusters in specific cancers such as breast cancer [18, 19], lung cancer [20], and gastric cancer [21]. We found two studies which were conducted in a population that included various cancer diagnoses with large sample sizes. In the study conducted by Morse, Cooper [22] among 1329 patients with various cancer diagnoses, eight symptom clusters were identified (i.e., physical and cognitive fatigue, respiratory, psychological, hormonal, chemotherapy-related toxicity, weight gain, gastrointestinal, and epithelial). Harris, Kober [23] found five symptom clusters, including psychological, gastrointestinal, weight gain, respiratory and hormonal clusters among more than 1000 outpatients with different cancers. In addition, researchers also pointed out there is no consensus on which symptoms occur in the population of cancer patients [24]. As such, the correlation between symptoms reported by cancer patients during treatment has not been fully addressed in the current literature. Furthermore, no study has been done previously to explore symptom clusters among cancer patients during chemotherapy in Vietnam, where people tend to skip their local health providers to go straight to the central hospitals for treatment. As the results, they have dealt with the bed-sharing problem (two or three patients share one bed). This issue not only caused so many inconveniences for the patients during their hospitalisation (from eating to resting) but also made the hospital become overloaded. Travelling a long way to receive treatment and facing persistent overload at hospitals have added burden to their treatments. Therefore, we conducted this study to assess the prevalence of symptoms that cancer patients experience during chemotherapy treatment and identify symptom clusters among such patients in the context of Vietnam, a low-middle-income country.

## Methods

### Study design

A cross-sectional design was conducted.

### Participants and settings

The study included 213 cancer patients from 3 large oncology hospitals in Vietnam (Vietnam National Cancer Institute, Bach Mai Hospital, and Hanoi Oncology Hospital). Vietnam National Cancer Institute: This is the largest cancer hospital in Vietnam, the hospital has 2400

beds and in charge of providing cancer treatments for patients from North and Middle of Vietnam (1700 patients per day on average). Bach Mai Hospital is one of the three largest general hospitals in Vietnam, its oncology unit has 300 beds and receives about 400 cancer patients per day on average. Hanoi Oncology Hospital is a public hospital in charge of providing cancer treatment for patients in Hanoi. The hospital has 680 beds and receives about 945 patients per day.

**Inclusion and exclusion criteria.** Patients were recruited if they were undergoing chemotherapy with any cancer diagnosis; had received at least one chemo-cycle, were 18–84 years old, and having a Karnofsky Performance Index $\geq$ 80. Patients were excluded if they had conditions that would prohibit completing the questionnaires, such as cognitive impairments or psychiatric disorders.

## Sample size calculation

Since the symptom cluster was identified by the correlation, we used the formula N $\geq$ 104 + m with "m" referring to the number of independent variables to calculate the sample size for the study [25]. There were 106 independent variables in the study (72 variables to measure 24 symptoms in 3 dimensions, 16 to measure 8 symptoms in 2 dimensions, and 18 to measure participant's characteristics). As such, the minimum sample size is 210. In this study, we sent out 250 invitations and received 213 responses. Therefore, the sample size of the study was 213.

## Procedures

Data were collected from 30[th] May to 28[th] July 2017 by 3 researchers using the convenience sampling method. Potential participants were approached and invited to the study in their ward. The researchers explained the purpose of the study. Participants who agreed to participate were informed of the procedures in detail and were provided with the study information sheet. Each participant signed the consent form. Afterwards, they were given the questionnaires to complete.

## Measurements

The Vietnamese version of the Memorial Symptom Assessment Scale (MSAS) ($\alpha$ = 0.79) was used to measure the symptom burden. The MSAS was translated into Vietnamese following the forward-backwards methods and had been validated by 5 experts in oncology. This is a self-reported questionnaire that measures 24 symptoms in 3 dimensions: frequency, severity, and distress; and eight symptoms were evaluated in terms of severity and distress (total of 32 symptoms). With each of the symptoms that the patients experienced, the frequency and severe dimensions were measured by a Likert scale from 1 to 4 (from "rarely" to "almost constantly" for the frequency and from "slight" to "very severe" for the severity). The distress dimension was a Likert scale from 0 to 4, with 0 referring to "not at all" and 4 referring to "very much". If a symptom is absent, each of the dimensions is scored as 0.

The MSAS has three sub-scales including Physical Symptom Subscale (PHYS), Psychological Symptom Subscale (PSYCH) and Global Distress Index (GDI). The PHYS is the average of the score for the 12 symptoms (lack of appetite, lack of energy, pain, feeling drowsy, constipation, dry mouth, nausea, vomiting, change in food taste, weight loss, feeling bloated, and dizziness). The PSYCH is the average of the score of the 6 symptoms (worrying, feeling sad, feeling nervous, difficulty sleeping, feeling irritable, and difficulty concentrating). The GDI is the average of the frequency of 4 psychological symptoms (feeling sad, worrying, feeling irritable, feeling nervous) and the distress associated with 6 physical symptoms (lack of appetite, lack of

energy, pain, feeling drowsy, constipation, dry mouth). The Total MSAS score (TMSAS) is the average of the symptom scores of all 32 symptoms [26]. The Cronbach's alpha was reported to be high in the current study. The overall scale had α = 0.92, with the psychological subscale α = 0.80 and the physical subscale α = 0.83.

## Ethical consideration

The study followed Helsinki's rules of ethics in medical research. Participants gave written informed consent to participate in the study. Only the first and corresponding authors had access to information that could identify participants during and after data collection. The study obtained ethical approval from the Human Subject Ethics Committee of Hanoi School of Public Health (Vietnam) and The Hong Kong Polytechnic University under reference number HSEARS20170428003.

## Data analysis

Data were analysed using SPSS 20.0. Descriptive statistics were used to report the prevalence and distress of the symptoms (number of participants and related percentage). It was also used to analyze the participant's demographic and clinical characteristics. Symptom clusters were identified by using Principal Component Analysis (PCA). To identify clinically meaningful symptom clusters, only symptoms with >20% prevalence and distress from "Quite a bit" to "very much" were selected for PCA [27]. There was no consensus about the cut-off value for the Pearson coefficient in cluster analysis. As such, a Pearson coefficient of > 0.3 was defined for a cluster [28]. The Kaise-Meyer-Olkin of the results (KMO) was greater than 0.80, which verified the sampling adequacy for the analysis, the Barlett's test of sphericity χ2 ranked from 2430.43 to 3651.06, p<0.001, indicated that correlation between symptoms was sufficiently large for PCA [29].

# Results

## Participants' characteristics

The majority of the participants were in the age group of 46 to 60 years old (48.8%; mean age was 53.1 years old). Most of the participants were female (128 patients, 60%). In the survey, the majority of patients were married (198, 88.7%) and being farmers or workers (80, 37.6%). Most of the participants had no comorbidities (n = 177, 83.1%), while for those who had, the most common comorbidity was hypertension (6.1%), followed by diabetes (3.3%). The majority of participants were diagnosed with breast cancer (59 patients, 27.7%). More than half of the patients had their tumor removed (121, 56.8%). The mean time from diagnosis was 13.7 (21.0) months (1–156). Texan-based chemotherapy regimen was the most common cancer treatment regimen among participants (102, 47.9%). Details of the demographic and clinical characteristics of the sample are shown in Table 1.

## Prevalence of symptoms

The common symptoms reported by participants were lack of appetite (65.3%); difficulty sleeping (62.9%), dry mouth (57.7%), numbness (60.1%), hair loss (43.2%), change in the way food tastes (40.8%), worrying (46%), feeling sad (41.8%), and lack of energy (44.1%).

The symptoms associated with distress (ranging from "quite a bit" to "very much") reported by more than 40% of participants were numbness (43.2%), difficulty sleeping (60.6%), and lack of appetite (44.1%).

**Table 1. Demographic and clinical characteristics of the patients.**

| | Patients (n = 213) | % |
|---|---|---|
| Age group | | |
| ≤ 45 | 54 | 25.4 |
| 46–60 | 104 | 48.8 |
| > 60 | 55 | 25.8 |
| Mean age (Mean ± SD) | 53.14 ± 11.3 (21–80) | |
| BMI (Mean ± SD) | 21.41 ± 2.53 (14.7–30.4) | |
| Marital status | | |
| Single | 13 | 6.1 |
| Married | 189 | 88.7 |
| Divorced/Widowed | 11 | 5.1 |
| Occupation | | |
| Unemployed | 30 | 14.1 |
| Retired | 51 | 23.9 |
| Laborer | 80 | 37.6 |
| Officer | 12 | 5.6 |
| Teacher | 11 | 5.2 |
| Others (Medical staff, Engineer, Freelancer) | 29 | 13.6 |
| Education | | |
| Primary school | 18 | 8.5 |
| High school or part of | 145 | 68.1 |
| College or part of | 28 | 13.1 |
| University or Higher | 22 | 10.3 |
| Comorbidity | | |
| None | 177 | 83.1 |
| Diabetes | 7 | 3.3 |
| Hypertension | 13 | 6.1 |
| Hypertension and diabetes | 3 | 1.4 |
| Degenerative spine | 5 | 2.3 |
| Others (Adipose hepatica, Gout, Dyslipidemia) | 8 | 3.8 |
| Cancer diagnosis | | |
| Breast cancer | 59 | 27.7 |
| Gynecologic cancer | 24 | 11.3 |
| Lung/Bronchial Cancer | 41 | 19.2 |
| Gastrointestinal Cancer | 38 | 17.8 |
| Non-Hopkin Lymphoma | 30 | 14.1 |
| Others (Nasopharyngeal cancer, Brain cancer, Amygdales cancer, Laryngeal cancer, tongue cancer, Nasopharynx cancer, Urinary system cancer) | 21 | 9.9 |
| Cancer stage | | |
| 1 | 28 | 13.1 |
| 2 | 66 | 31.0 |
| 3 | 62 | 29.1 |
| 4 | 57 | 26.8 |
| Surgical debulking | | |
| Optimal | 121 | 56.8 |
| Sub-optimal | 20 | 9.4 |
| None | 72 | 33.8 |

(*Continued*)

**Table 1.** (Continued)

| | Patients (n = 213) | % |
|---|---|---|
| Chemotherapy Regimen | | |
| Taxane based Chemotherapy | 102 | 47.9 |
| Cyclophosphamide and doxorubicin based Chemotherapy | 50 | 23.5 |
| Oxaliptatin based Chemotherapy | 23 | 10.8 |
| Gemcitabine based Chemotherapy | 18 | 8.5 |
| Others (Vinorelbine, Rituximab, Trastuzumab) | 20 | 9.4 |

The symptoms associated with severe (ranging from "moderate" to "very severe") were Lack of energy, dry mouth, numbness, difficulty sleeping, feeling bloated, problems with urination, vomiting, worrying, problems with sexual interests or activity, lack of appetite, feeling irritable, change in the way food tastes, hair loss, constipation, and "I don't look like myself". Details of the prevalence, distress and severity of symptoms reported by participants are shown in Table 2.

Among 3 subscales of MSAS, the Psychological subscale had the highest score (1.12 ± 0.78) (Table 3).

## Symptom clusters

Four symptom clusters were identified among participants. Most of the identified symptom clusters included physical symptoms. *Difficult concentration*, *pain*, *cough*, and *dizziness* contributed to the first SC. The second one included *lack of energy*, *numbness*, *change the food taste*, and *lack of appetite*. *Dry mouth*, *nausea*, *feeling bloated*, *problems with urination*, *hair loss*, and *constipation* made up for the third SC. The final SC consisted of psychological symptoms which were *feeling nervous*, *feeling drowsy*, *feeling sad*, *worrying*, *problems with sexual interests or activity and difficulty sleeping* (Table 4).

## Discussion

### Prevalence of symptoms

The first aim of this study was to assess the prevalence of symptoms that patients experienced during chemotherapy. The most prevalent symptoms (reported by more than 40% of participants) were *lack of appetite*, *difficulty sleeping*, *dry mouth*, *numbness*, *hair loss*, *change in the way food tastes*, *and lack of energy*. This was similar to findings from the study of Supaporn Chongkham-ang et al. (2018) performed on 20 breast cancer patients who were treated with chemotherapy. It included five prevalence symptoms: hair loss, lack of energy, lack of appetite, change in the way food tastes, and nausea [30]. The study conducted by Mohammad Al Qadire et al. (2023) showed that in 393 cancer patients, symptoms related to tiredness, lack of energy, and irritability were the most common [31]. In line with our results, Carolyn S Harris et al. (2022) found that lack of energy was the most prevalent symptom in 1329 outpatients with cancer during chemotherapy [32]. As such, our study indicated a similarity of common symptoms experienced by cancer patients in Vietnam with those in other countries.

The results indicated that the most prevalent symptoms were not the most distressing. Therefore, our findings revealed that *difficulty sleeping, loss of appetite, and numbness* were among the most distressing symptoms for cancer patients in Vietnam undergoing chemotherapy. This was similar to the study of Mohammad AI-Qadir et al. (2023), who indicated that the most distressing symptom was a lack of appetite among 393 participants (31). By contrast, in

**Table 2. Prevalence and distress of the symptom experienced by participants (n = 213).**

| Symptom | Prevalence % | Distress % (from "Quite a bit to 'Very much) | Severity* Mean (SD) |
|---|---|---|---|
| Difficult concentration | 84 (39.4%) | 45 (21.1%) | 1.8 (0.94) |
| Pain | 71 (33.3%) | 48 (22.5%) | 1.9 (1.01) |
| Lack of Energy | **94 (44.1%)** | 65 (30.5%) | **2.14 (1.21)** |
| Cough | 82 (38.5%) | 49 (23%) | 1.95 (0.98) |
| Feeling nervous | 61 (28.6%) | 43 (20.2%) | 1.89 (0.93) |
| Dry mouth | **123 (57.7%)** | 76 (35.%) | **2.08 (1.03)** |
| Nause | 74 (34.7%) | 50 (23.5%) | 1.89 (0.99) |
| Feeling drowsy | 72 (33.8%) | 43 (20.2%) | 1.79 (8.85) |
| Numbness | **128 (60.1%)** | 92 (43.2%) | **2.16 (1.04)** |
| Difficulty sleeping | **134 (62.9%)** | 129 (60.6%) | **2.2 (0.9)** |
| Feeling Bloated | 66 (31%) | 48 (22.5%) | **2.21 (1.37)** |
| Problems with Urination | 57 (26.8%) | 43 (20.2%) | **2.01 (0.97)** |
| Vomit | 48 (22.5%) | 35 (16.4%) | **2.1 (1.11)** |
| Short of breath | 65 (30.5%) | 40 (18.8%) | 1.83 (0.94) |
| Diarrhea | 38 (17.8%) | 24 (11.3%) | 1.65 (0.86) |
| Feeling sad | **89 (41.8%)** | 64 (30%) | 1.98 (1.01) |
| Sweats | 63 (29.6%) | 37 (17.4%) | 1.83 (0.98) |
| Worrying | **98 (46%)** | 67 (31.5%) | **2.03 (0.93)** |
| Problems with sexual interests or activity | 74 (34.7%) | 51 (23.9%) | **2.5 (1.18)** |
| Itching | 52 (24.4%) | 32 (15%) | 1.87 (1.04) |
| Lack of appetite | **139 (65.3%)** | 94 (44.1%) | **2.1 (1.03)** |
| Dizziness | **91 (42.7%)** | 65 (30.5%) | 1.94 (0.95) |
| Difficulty swallowing | 56 (26.3%) | 39 (18.3%) | 1.95 (0.95) |
| Feeling irritable | 84 (39.4%) | 63 (29.6%) | **2.1 (0.98)** |
| Mouth sores | 34 (16.0%) | 33 (15.5%) | 1.7 (0.89) |
| Change in the way food tastes | **87 (40.8%)** | 74 (34.7%) | **2.01 (0.82)** |
| Weigh loss | 36 (16.9%) | 26 (12.2%) | 1.58 (0.7) |
| Hair loss | **92 (43.2%)** | 82 (38.5%) | **2.63 (1.17)** |
| Constipation | 66 (31%) | 56 (26.3%) | **2.1 (0.88)** |
| Swelling of arms or legs | 14 (6.6%) | 13 (6.1%) | 1.61 (0.89) |
| I don't look like myself | 32 (15%) | 31 (14.6%) | **2.01 (0.94)** |
| Changes in skin | 29 (16.3%) | 24 (11.3%) | 1.84 (0.86) |

* severity items ranged from one (slight) to four (very severe)

the study by Supaporn Chongkham-ang et al. (2018), hair loss was the symptom that caused the most distress, although lack of appetite was the most common symptom (30). In addition, Carolyn S Harris et al. (2022) indicated that "*I don't look like me*" was a distressing symptom for cancer patients [32]. A survey of 232 gynaecological cancer patients during chemotherapy

**Table 3. The score of MSAS's sub-scales.**

| Sub-scales | Mean ± SD (min-max) |
|---|---|
| PHYS | 0.96 ± 0.67 (0–2.96) |
| PSYCH | **1.12 ± 0.78 (0–3.44)** |
| GDI | 1.08 ± 0.81 (0–3.6) |
| TMSAS | 0.91 ± 0.57 (0–2.65) |

**Table 4. Identified symptom clusters among participants.**

|  | Symptom components | r |
|---|---|---|
| Symptom cluster 1 | Difficult concentration | 0.7 |
|  | Pain | 0.51 |
|  | Cough | 0.57 |
|  | Dizziness | 0.64 |
| Symptom cluster 2 | Lack of Energy | 0.6 |
|  | Numbness | 0.7 |
|  | Change food taste | 0.7 |
|  | Lack of appetite | 0.7 |
| Symptom cluster 3 | Dry mouth | 0.5 |
|  | Nausea | 0.57 |
|  | Feeling Bloated | 0.58 |
|  | Problems with urination | 0.56 |
|  | Hair loss | 0.48 |
|  | Constipation | 0.7 |
| Symptom cluster 4 | Feeling nervous | 0.46 |
|  | Feeling drowsy | 0.5 |
|  | Feeling sad | 0.4 |
|  | Worrying | 0.4 |
|  | Problems with sexual interests or activity | 0.4 |
|  | Difficulty sleeping | 0.7 |
| **KMO** | 0.86 | |
| $\chi^2$ | 2435.52 | |
| **p** | <0.001 | |

also indicated lack of energy, hair loss, and "*I look not like myself*" were the most common, severe, and distress symptoms of subjects [33]. The differences between the existing studies could be explained by cultural factors. This could be because cultural differences influence the response to symptoms among cancer patients [34]. Therefore, there were different results of symptoms causing distress to those who underwent chemotherapy.

Among 3 sub-scales of MSAS, the Psychological subscale has the highest score (1.12 ± 0.78), which revealed oncology patients under chemotherapy suffer a greater burden of psychological symptoms (*worrying*, *feeling sad*, *feeling nervous*, *difficulty sleeping*, *feeling irritable*, *and difficulty concentrating*) than other symptoms. Other studies also reported similar results. Research conducted among 4500 patients diagnosed with the 14 most prevalent types of cancer showed a significant psychological distress proportion ranging from 29 to 43% [35]. A recent survey also indicated about 25% of cancer patients experienced psychological distress that partly led to quality of life reduction [36]. These results indicate the need for psychological care and support during chemotherapy for cancer patients.

## Symptom cluster

To the best of our knowledge, this study is the first to identify symptoms cluster in cancer patients undergoing chemotherapy in Vietnam–a low-middle-income country. We identified four symptom clusters among participants.

The first symptom cluster was *pain, cough, dizziness, and difficult concentration*. The correlation among these symptoms was moderate (*r* = 0.51–0.7). This finding was similar to studies conducted Atay, Conk and Bahar [27] and Yeh, Chiang [37], which found that *cough* and

*dizziness* fell into the same cluster. In addition, Chen and Tseng [13] and also found *pain*, *difficult concentration and dizziness* appeared in the "sickness symptom cluster" which was mediated by proinflammatory cytokines [37, 38]. In this study, pain may contribute to dizziness and difficulty concentrating.

The second symptom cluster was *lack of energy, numbness, change food taste, and lack of appetite*. This symptom cluster was reported in other studies among different cancer populations [27, 38]. Perhaps the sentinel symptom of this cluster was lack of appetite which related to change food taster and led to lack of energy and numbness.

The third symptom cluster we found was *dry mouth*, *nausea*, *feeling bloated*, *problem with urination*, *hair loss*, *and constipation*. The symptom clusters related to hair loss and gastrointestinal symptoms could be the result of side effects during chemotherapy [39]. Doxorubicin, cyclophosphamide, and paclitaxel were included in chemo components and were reported to partly cause the symptom clusters [40]. This finding was in accordance with the results of the symptom cluster found among Taiwanese patients with various cancer diagnoses [13].

The last symptom cluster that appeared in the sample was included psychological symptoms such as *feeling nervous*, *feeling drowsy*, *feeling sad*, *worrying*, *problems with sexual interests or acitivity*, *and difficulty sleeping*. Other studies reported the similar findings [9, 13, 27, 41, 42]. The bidirectional relationship between sleep disturbances and depressive and anxious symptoms (*feeling sad*, *worrying*, *feeling nervou*s) has been well documented in the literature [43, 44]. In this cluster, the core symptom that defines the cluster with the highest correlation is difficulty sleeping. Sleep disturbances lead to the development of feeling sad, worrying, feeling nervous, and make the patients have problems with sexual interests or acitivites. In addition, other studies indicated that psychology symptom clusters were the most prevalent and the most distressed ones [45]. This result explained the reason for the highest score of psychological subscale when using MSAS to measure symptoms in the current study.

Our study is characterised by several strengths, including the use of large sample size, multi-site data collection, sample diagnostic heterogeneity, and measuring a wide range of symptoms using a validated measurement. However, due to limitations of cross-sectional design, it was impossible to point out which symptom in each cluster occurred first nor how they influenced each other, or how these symptoms/symptom clusters changed over the course of the illness. Furthermore, the data was collected before the emergence of the COVID-19 pandemic. COVID-19 adversely affects cancer patients because they are immunocompromised [46]. As such, a repeating study is recommended to understand the change in the symptom burden experienced by cancer patients after the pandemic.

Findings from the study indicated cancer patients undergoing chemotherapy endure a wide range of symptoms. The most prevalent symptoms were lack of appetite, difficulty sleeping, dry mouth, numbness, hair loss, change in the way food tastes, and lack of energy. Four symptom clusters have been identified including different physical and psychological symptoms among participants. The findings shown that physical and psychological symptom clusters of those patients are inevitable. Symptoms could co-occur due to different pathophysiological reasons and mechanisms [47, 48], but they have an extremely negative impact on cancer patients. That could cause oncology patients to endure discomforts, the physical function would change, and their mental and emotional abilities would be reduced which reduce patient's quality of life [49]. This current study highlights the burden of chemotherapy treatment faced by patients, providing valuable information to assist oncologists/clinicians in treatment planning, management of side effects, and caring for these patients. In addition, our study calls for further comprehensively assessing symptoms among patients undergoing chemotherapy regardless of durations, types, and toxicities of chemotherapy.

## Acknowledgments

We would like to thank all the participants for their voluntary contribution to the study.

## Author Contributions

**Conceptualization:** Quang Vinh Bui.

**Data curation:** Quyen Thi Le Le, Anh Thi Hong Nguyen.

**Formal analysis:** Huong Thi Xuan Hoang, Quyen Thi Le Le, Vi Pham Nhat Do, Anh Thi Hong Nguyen, Quang Vinh Bui.

**Investigation:** Quyen Thi Le Le, Quang Vinh Bui.

**Methodology:** Huong Thi Xuan Hoang, Anh Thi Hong Nguyen.

**Project administration:** Huong Thi Xuan Hoang.

**Supervision:** Huong Thi Xuan Hoang.

**Validation:** Huong Thi Xuan Hoang.

**Writing – original draft:** Huong Thi Xuan Hoang, Quyen Thi Le Le, Vi Pham Nhat Do, Anh Thi Hong Nguyen, Quang Vinh Bui.

**Writing – review & editing:** Huong Thi Xuan Hoang.

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
