## [Decision Letter · Decision Letter 0]

6 Mar 2024

PONE-D-23-41096Identifying Symptom Cluster in Cancer patients undergoing Chemotherapy in Vietnam: a cross-sectional studyPLOS ONE

Dear Dr. Hoang,

Thank you for submitting your manuscript to PLOS ONE. After careful consideration, we feel that it has merit but does not fully meet PLOS ONE’s publication criteria as it currently stands. Therefore, we invite you to submit a revised version of the manuscript that addresses the points raised during the review process.

We look forward to receiving your revised manuscript.

Kind regards,

Alison Wang

Academic Editor

PLOS ONE

2. We noted in your submission details that a portion of your manuscript may have been presented or published elsewhere. [The data related to participants' characteristics have been published in a paper (DOI: 10.1007/s11325-019-01839-x)

However, the previous paper was done with a different aim. We just took the data related to the characteristics of participants to include in this manuscript since the study was done in the same sample with the previous paper.

The first author os the previous paper is also the corressponding author of the current manuscript.] Please clarify whether this publication was peer-reviewed and formally published. If this work was previously peer-reviewed and published, in the cover letter please provide the reason that this work does not constitute dual publication and should be included in the current manuscript.

3. In the online submission form, you indicated that [Data are available upon acceptance of the corressponding author]. 

Reviewers' comments:

Reviewer's Responses to Questions

**Comments to the Author**

1. Is the manuscript technically sound, and do the data support the conclusions?

Reviewer #1: Yes

Reviewer #2: Yes

Reviewer #3: Partly

2. Has the statistical analysis been performed appropriately and rigorously? 

Reviewer #1: I Don't Know

Reviewer #2: Yes

Reviewer #3: Yes

3. Have the authors made all data underlying the findings in their manuscript fully available?

Reviewer #1: Yes

Reviewer #2: No

Reviewer #3: Yes

4. Is the manuscript presented in an intelligible fashion and written in standard English?

Reviewer #1: Yes

Reviewer #2: No

Reviewer #3: Yes

5. Review Comments to the Author

**Reviewer #1: **Thank you for the opportunity to read this manuscript.

This study aimed to assess the prevalence of symptoms experienced by cancer patients during chemotherapy and to identify symptom clusters during chemotherapy in such patients. The research topic is interesting and the manuscript is clearly written. However, there are several issues that need to be addressed. My suggestions are as follows:

Introduction

- Line 60: There should be more than one previous study investigating the symptom clusters in patients receiving chemotherapy. Please elaborate on results of previous studies to justify why this study is needed.

Methods

- Line 77: Please provide more information about the settings of Vietnam to allow reader to understand how representative the sample is. How many large oncology hospitals are there? How many patients do they serve?

- Line 79: Why recruit 20-84 years old? What about 18-19 years old?

- Line 66: Please elaborate on how bed-sharing issues affect the symptom cluster?

- Please provide the ethical approval reference no.

Discussion

- Please provide details on the implications of the study results, given that patients received varying durations, types, and toxicities of chemotherapy.

**Reviewer #2**: The premise of this study to identify symptom clusters among people undergoing chemotherapy for cancer in Vietnam is significant. Defining the clusters using principal component analysis was appropriate. Several areas warrant consideration, as follows:

1. The statement in the background (page 3, 59-61) that most of the early research measured only 13-15 symptoms is untrue. One of the seminal studies, among others, identified 25 commonly occurring symptoms (PMID: 24797450).

2. The methods section includes a description of the MSAS instrument, including the 3 dimensions the instrument measures. The manuscript reports occurrence and distress related to each symptom but not severity. Either include or justify why this dimension was excluded.

3. It would be helpful develop a table that includes all dimensions being measured and assessed for symptom clusters (e.g., PMID 34560709).

4. Statements such as the sentence on page 12 (lines 253-254) (i.e., “This proves that….”) is inaccurate to state based on one study. This should be rephrased to “The findings suggest….” Or something similar.

5. Overall, the level of grammatical errors and to a lesser degree, spelling errors detracted from the merits of this manuscript. Assistance from an English-speaking scientist is highly recommended.

Overall, this investigation is significant, and merits being reported in the literature. Addressing the above points would strengthen this manuscript tremendously.

**Reviewer #3: **The manuscript shows the prevalence of symptoms in oncology patients using the MSAS scale in Vietnam. The article is interesting but I have several considerations to mention.

Line 59-61. It cannot be stated that there are no studies assessing this prevalence of 32 symptoms because there are (e.g. doi: 10.3390/ijerph20031708).

There is no sample size calculation.

The inclusion criteria are not clear: as you may know the MSAS assesses the symptoms perceived during the week prior to the chemotherapy session, therefore, patients who are going to receive their first cycle of chemotherapy should not be included since the symptoms they may suffer will not be due to chemotherapy and may represent a bias.

It's been 7 years since the data were collected, are they still representative? haven't they changed even with a pandemic in the middle?

Line 95: "the symptom was measured using a Likert scale from 0 to 4 with a higher score" that is not true, the categories of frequency and intensity are measured from 1 to 4 while distress is measured from 0 to 4 (having 5 possible options).

Is the MSAS scale in Vietnamese validated in oncological patients? Since the only reference I have found is the short version and in HIV patients (10.1080/09540121.2021.1922577)

Line 107-109: these values have to be the reference values in the language in which it is to be used, not in the original.

When were the data collected? Where were they collected? Were they self-completed? Did you go through the MSAS on your own? Did someone help you to fill them in?

Results:

Has any stratified analysis been performed in relation to the sex or age of the participants, even taking into account the type of cancer?

Discussion:

In the introduction they comment that there are no studies with large samples, however, in the discussion reference 23 and 24 contradict this information.

Again, the reference doi: 10.3390/ijerph20031708 should be cited in the text.

Line 209: I am not very clear about the symptom cluster done on the basis of what, prevalence? Could you explain it better, please.

Line 238 “feeling nervour”; line 239 “acitivites.” please correct these words.

6. PLOS authors have the option to publish the peer review history of their article (what does this mean?). If published, this will include your full peer review and any attached files.

Reviewer #1: No

Reviewer #2: No

Reviewer #3: No

---

## [Author Response · Author response to Decision Letter 0]

10 May 2024

Dear Reviewers

On behalf of the research team, I would like to thank you for allowing us to submit a revised draft of the manuscript “Identifying Symptom Cluster in Cancer patients undergoing Chemotherapy in Vietnam: a cross-sectional study”. We appreciate the time and effort you dedicated to providing feedback on our manuscript and are grateful for the insightful comments that helped us improve our paper. We have incorporated most of the suggestions made by the reviewers. Those changes are highlighted within the manuscript. Please see below for a point-by-point response to the reviewers’ comments and concerns. All page numbers refer to the revised manuscript file with tracked changes.

Reviewers' Comments to the Authors:

 Reviewer 1

Line 60: There should be more than one previous study investigating the symptom clusters in patients receiving chemotherapy. Please elaborate on results of previous studies to justify why this study is needed.

Thank you for pointing this out. The reviewers are correct and we have revised as following (Line 62 to 72)

In addition, previous studies explored symptom clusters in specific cancers such as breast cancer (1, 2), lung cancer (3), and gastric cancer (4). We found two studies which were conducted in a population that included various cancer diagnoses with large sample sizes. In the study conducted by Morse, Cooper (5) among 1329 patients with various cancer diagnoses, eight symptom clusters were identified (i.e., physical and cognitive fatigue, respiratory, psychological, hormonal, chemotherapy-related toxicity, weight gain, gastrointestinal, and epithelial). Harris, Kober (6) found five symptom clusters, including psychological, gastrointestinal, weight gain, respiratory and hormonal clusters among more than 1000 outpatients with different cancers. In addition, researchers also pointed out there is no consensus on which symptoms occur in the population of cancer patients (7). As such, the correlation between symptoms reported by cancer patients during treatment has not been fully addressed in the current literature.

Line 66: Please elaborate on how bed-sharing issues affect the symptom cluster?

As suggested, we have revised.

Line 73-78

Furthermore, no study has been done previously to explore symptom clusters among cancer patients during chemotherapy in Vietnam, where people tend to skip their local health providers to go straight to the central hospitals for treatment. As the results, they have dealt with the bed-sharing problem (two or three patients share one bed). This issue not only caused so many inconveniences for the patients during their hospitalisation (from eating to resting) but also made the hospital become overloaded.

Please provide the ethical approval reference no.

We have inserted the ethical approval reference (Line 157-163)

The study obtained ethical approval from the Human Subject Ethics Committee of Hanoi School of Public Health (Vietnam) and The Hong Kong Polytechnic University under reference number HSEARS20170428003

Line 77: Please provide more information about the settings of Vietnam to allow the reader to understand how representative the sample is. How many large oncology hospitals are there? How many patients do they serve?

Thank you for these comments, we have added details of the settings in the “Participants and settings” session (line 112-121)

The study included 213 cancer patients from 3 large oncology hospitals in Vietnam (Vietnam National Cancer Institute, Bach Mai Hospital, and Hanoi Oncology Hospital). Vietnam National Cancer Institute: This is the largest cancer hospital in Vietnam, the hospital has 2400 beds and in charge of providing cancer treatments for patients from North and Middle of Vietnam (1700 patients per day on average). Bach Mai Hospital is one of the three largest general hospitals in Vietnam, its oncology unit has 300 beds and receives about 400 cancer patients per day on average. Hanoi Oncology Hospital is a public hospital in charge of providing cancer treatment for patients in Hanoi. The hospital has 680 beds and receives about 945 patients per day. 

Line 79: Why recruit 20-84 years old? What about 18-19 years old?

Thank you for pointing out this major issue of the manuscript. In fact, we recruited participants from 18-80 years old. However, none of the participants were in the age group of 18-19. Thanks to your comments we have revised the criteria. (line 123-124)

Patients were recruited if they were undergoing chemotherapy with any cancer diagnosis, had received at least one chemo-cycle, were 18–84 years old

Please provide details on the implications of the study results, given that patients received varying durations, types, and toxicities of chemotherapy

Thank to your comments, we have revised and highlighted the implication of the study

(Line 449-453)

This current study highlights the burden of chemotherapy treatment faced by patients, providing valuable information to assist oncologists/clinicians in treatment planning, management of side effects, and caring for these patients. In addition, our study calls for further comprehensively assessing symptoms among patients undergoing chemotherapy regardless of durations, types, and toxicities of chemotherapy.

Reviewer 2

The statement in the background (page 3, 59-61) that most of the early research measured only 13-15 symptoms is untrue. One of the seminal studies, among others, identified 25 commonly occurring symptoms (PMID: 24797450).

Thank you for your comments. We have done the literature review again and revised as following: (line 69-71)

However, the majority of the earlier research was conducted in a small sample size, and some studies only measured only 13 to 15 symptoms, while cancer patients can experience up to 32 symptoms (8).

The methods section includes a description of the MSAS instrument, including the 3 dimensions the instrument measures. The manuscript reports occurrence and distress related to each symptom but not severity. Either include or justify why this dimension was excluded.

Thank you for this comment. We have added a column to report the severity of the symptoms. Please see table 2 (page 240-265)

Statements such as the sentence on page 12 (lines 253-254) (i.e., “This proves that….”) is inaccurate to state based on one study. This should be rephrased to “The findings suggest….” Or something similar.

As suggested, we have paraphrased these statements

(Line 444-445)

The findings shown that physical and psychological symptom clusters of those patients are inevitable.

Overall, the level of grammatical errors and to a lesser degree, spelling errors detracted from the merits of this manuscript. Assistance from an English-speaking scientist is highly recommended

We acknowledge the limitation of being non-native English-speaking scientists. Since the project received no funds so we have invited an English-speaking scientist from the University of Queensland to check the manuscript before re-submitting. The grammatical errors and typos have been revised.

Review 3

Line 59-61. It cannot be stated that there are no studies assessing this prevalence of 32 symptoms because there are (e.g. doi: 10.3390/ijerph20031708).

While we appreciate the reviewer’s feedback, we respectfully disagree because we haven’t stated studies assessing this prevalence of 32 symptoms in the previous manuscript. Maybe our writing confused you. Sorry for that. We meant “no study has been done previously to explore symptom clusters among cancer patients during chemotherapy in Vietnam where people tend to skip their local health providers to go straight to the central hospitals for treatment” (line 82-84).

There is no sample size calculation.

Thank you for pointing out the major issue of the manuscript. We have added the sample size calculation

Line 127-134

Sample size calculation

Since the symptom cluster was identified by the correlation, we used the formula N ≥104 + m with “m” referring to the number of independent variables to calculate the sample size for the study (9). There were 106 independent variables in the study (72 variables to measure 24 symptoms in 3 dimensions, 16 to measure 8 symptoms in 2 dimensions, and 18 to measure participant’s characteristics). As such, the minimum sample size is 210. In this study, we sent out 250 invitations and received 213 responses. Therefore, the sample size of the study was 213.

The inclusion criteria are not clear: as you may know the MSAS assesses the symptoms perceived during the week prior to the chemotherapy session, therefore, patients who are going to receive their first cycle of chemotherapy should not be included since the symptoms they may suffer will not be due to chemotherapy and may represent a bias

The reviewer was right and we have edited the inclusion criteria to make it clear

(Line 122-126)

Inclusion and exclusion criteria: 

Patients were recruited if they were undergoing chemotherapy with any cancer diagnosis; had received at least one chemo-cycle, 18–84 years old, and having a Karnofsky Performance Index ≥ 80.

It's been 7 years since the data were collected, are they still representative? haven't they changed even with a pandemic in the middle?

Thank you for your valuable questions. This is the limitation of the study, we have added to the discussion

Line 390-393

Furthermore, the data was collected before the emergence of the COVID-19 pandemic. COVID-19 adversely affects cancer patients because they are immunocompromised (10). As such, a repeating study is recommended to understand the change in the symptom burden experienced by cancer patients after the pandemic.

Line 95: "the symptom was measured using a Likert scale from 0 to 4 with a higher score" that is not true, the categories of frequency and intensity are measured from 1 to 4 while distress is measured from 0 to 4 (having 5 possible options).

Is the MSAS scale in Vietnamese validated in oncological patients? Since the only reference I have found is the short version and in HIV patients (10.1080/09540121.2021.1922577)

We think this is an excellent correction. We have revised in the manuscript

Line 144-164

The Vietnamese version of the Memorial Symptom Assessment Scale (MSAS) (α = 0.79) was used to measure the symptom burden. The MSAS was translated into Vietnamese following the forward-backward methods and had been validated by 5 experts in oncology. This is a self-reported questionnaire that measures 24 symptoms in 3 dimensions: frequency, severity, and distress; and eight symptoms were evaluated in terms of severity and distress (total of 32 symptoms). With each of the symptoms that the patients experienced, the frequency and severe dimensions were measured by a Likert scale from 1 to 4 (from “rarely” to “almost constantly” for the frequency and from “slight” to “very severe” for the severity). The distress dimension was a Likert scale from 0 to 4 with 0 referring to “not at all” and 4 refers to “very much”. If a symptom is absent, each of the dimensions is scored as 0. 

Has any stratified analysis been performed in relation to the sex or age of the participants, even taking into account the type of cancer?

We conducted this study to assess the prevalence of symptoms that cancer patients experience during chemotherapy treatment and identify symptom clusters among such patients in the context of Vietnam. So no stratified analysis was performed in relation to sex or age and cancer type was taken.

In the introduction they comment that there are no studies with large samples, however, in the discussion reference 23 and 24 contradict this information.

Again, the reference doi: 10.3390/ijerph20031708 should be cited in the text.

Thank you for suggesting. We have revised.

(Line 73-80)

We found two studies which were conducted in a population that included various cancer diagonises with large sample size . In the study conducted by Morse, Cooper (5) among 1329 patients with various cancer diagnoses, eight symptom clusters were identified (i.e., physical and cognitive fatigue, respiratory, psychological, hormonal, chemotherapy-related toxicity, weight gain, gastrointestinal, and epithelial). Harris, Kober (6) found five symptom clusters including psychological, gastrointestinal, weight gain, respiratory and hormonal clusters among more than 1000 outpatients with different cancers. In addition, researchers also pointed out there is no consensus on which symptoms occur in the population of cancer patients (7).

Line 209: I am not very clear about the symptom cluster done on the basis of what, prevalence? Could you explain it better, please.

In according to the definition “A symptom cluster has been defined as at least two symptoms that occurred and related to each other at a certain time”, the symptom cluster was identified based on the prevalence of the symptom.

Line 238 “feeling nervous”; line 239 “acitivites.” please correct these words.

We have revised the typo. Thank you and sorry for these mistakes.

References

1. Browall M, Brandberg Y, Nasic S, Rydberg P, Bergh J, Rydén A, et al. A prospective exploration of symptom burden clusters in women with breast cancer during chemotherapy treatment. Supportive Care in Cancer. 2017;25(5):1423-9.

2. Wiggenraad F, Bolam KA, Mijwel S, van der Wall E, Wengström Y, Altena R. Long-Term Favorable Effects of Physical Exercise on Burdensome Symptoms in the OptiTrain Breast Cancer Randomized Controlled Trial. Integrative Cancer Therapies. 2020;19:1534735420905003.

3. Teng L, Zhou Z, Yang Y, Sun J, Dong Y, Zhu M, Wang T. Identifying central symptom clusters and correlates in patients with lung cancer post-chemotherapy: A network analysis. Asia-Pacific Journal of Oncology Nursing. 2024;11(4):100383.

4. Fu L, Feng X, Jin Y, Lu Z, Li R, Xu W, et al. Symptom Clusters and Quality of Life in Gastric Cancer Patients Receiving Chemotherapy. Journal of Pain and Symptom Management. 2022;63(2):230-43.

5. Morse L, Cooper BA, Ritchie CS, Wong ML, Kober KM, Harris C, et al. Stability and consistency of symptom clusters in younger versus older patients receiving chemotherapy. BMC Geriatrics. 2024;24(1):164.

6. Harris CS, Kober KM, Cooper B, Conley YP, Dhruva AA, Hammer MJ, et al. Symptom clusters in outpatients with cancer using different dimensions of the symptom experience. Support Care Cancer. 2022;30(8):6889-99.

7. Llamas-Ramos I, Alvarado-Omenat JJ, Rodrigo-Reguilón M, Llamas-Ramos R. Quality of Life and Side Effects Management in Cancer Treatment-A Cross Sectional Study. Int J Environ Res Public Health. 2023;20(3).

8. Lobchuk MM. The memorial symptom assessment scale: modified for use in understanding family caregivers' perceptions of cancer patients' symptom experiences. J Pain Symptom Manage. 2003;26(1):644-54.

9. Kupper LL, Hafner KB. How Appropriate Are Popular Sample Size Formulas? The American Statistician. 1989;43(2):101-5.

10. Linjawi M, Shakoor H, Hilary S, Ali HI, Al-Dhaheri AS, Ismail LC, et al. Cancer Patients during COVID-19 Pandemic: A Mini-Review. Healthcare (Basel). 2023;11(2).

---

## [Decision Letter · Decision Letter 1]

10 Jul 2024

Identifying Symptom Cluster in Cancer patients undergoing Chemotherapy in Vietnam: a cross-sectional study

PONE-D-23-41096R1

Dear Dr. Huong Thi Xuan Hoang,

We’re pleased to inform you that your manuscript has been judged scientifically suitable for publication and will be formally accepted for publication once it meets all outstanding technical requirements.

Kind regards,

Alison Wang

Academic Editor

PLOS ONE

Additional Editor Comments (optional):

Reviewers' comments:

Reviewer's Responses to Questions

**Comments to the Author**

1. If the authors have adequately addressed your comments raised in a previous round of review and you feel that this manuscript is now acceptable for publication, you may indicate that here to bypass the “Comments to the Author” section, enter your conflict of interest statement in the “Confidential to Editor” section, and submit your "Accept" recommendation.

Reviewer #1: All comments have been addressed

Reviewer #2: All comments have been addressed

2. Is the manuscript technically sound, and do the data support the conclusions?

Reviewer #1: Yes

Reviewer #2: (No Response)

3. Has the statistical analysis been performed appropriately and rigorously? 

Reviewer #1: Yes

Reviewer #2: (No Response)

4. Have the authors made all data underlying the findings in their manuscript fully available?

Reviewer #1: Yes

Reviewer #2: (No Response)

5. Is the manuscript presented in an intelligible fashion and written in standard English?

Reviewer #1: Yes

Reviewer #2: (No Response)

6. Review Comments to the Author

Reviewer #1: Thanks for the revision! It is well-written, and all of my comments were addressed. Thank you very much!

Reviewer #2: (No Response)

7. PLOS authors have the option to publish the peer review history of their article (what does this mean?). If published, this will include your full peer review and any attached files.

Reviewer #1: No

Reviewer #2: No

---

## [Editor Report · Acceptance letter]

20 Jul 2024

PONE-D-23-41096R1 

PLOS ONE

Dear Dr. Hoang, 

I'm pleased to inform you that your manuscript has been deemed suitable for publication in PLOS ONE. Congratulations! Your manuscript is now being handed over to our production team.

Kind regards, 

on behalf of

Dr. Tao (Alison) Wang 

Academic Editor

PLOS ONE